# TwoTorials: A Remote Cooperative Tutorial System for 3D Design Software

Sultan A. Alharthi[1α], Ben Lafreniere[2α], Tovi Grossman[3μα], George Fitzmaurice[4α]

[α]Autodesk Research and [μ]University of Toronto

## ABSTRACT

Step-by-step tutorials have emerged as a key means for learning complex software, but they are typically designed for individuals learning independently. In contrast, cooperative learning, where learners can help each other as they work, is a fundamental pedagogical technique with many established benefits. To extend these benefits to learning 3D-design software, this work investigates the design of remote cooperative software tutorial systems. We first conduct an observational study of pairs of participants working on 3D-design tutorials, which reveals a range of potential benefits, challenges, and strategies for cooperation. Our findings inform the design of *TwoTorials*, a cooperative step-by-step tutorial system that helps pairs of remote users establish shared 3D context, maintain awareness of each other's activities, and coordinate their efforts. A user study reveals several benefits to this approach, including enhanced cooperation between learners, reduced effort and mental demand, increased awareness of peer activities, and higher subjective engagement with the tutorial.

**Keywords**: Software learning, 3D modeling, remote learning

## 1 INTRODUCTION

Users starting out in 3D design software face a range of learnability challenges [36], which have motivated the development of a variety of innovative software learning systems (e.g., [13, 36, 51, 59, 78]). In particular, step-by-step tutorials have emerged as a key means for learning complex software, and tutorials of this type exist for nearly all popular applications. In some ways, these tutorials replicate the experience of working on non-trivial projects using the software, with the tutorial providing a clear goal and scaffolding the user's skills and abilities [56]. However, this format of tutorials is primarily designed for individuals learning independently, so users cannot benefit from over-the-shoulder learning [88] and other advantages that come from learning alongside other people, such as occurs in workplace settings. This is unfortunate, because education research has established cooperative learning as a fundamental pedagogical technique [22] with benefits in terms of learner motivation [18, 75], retention [5, 80], and effective knowledge gain and transfer [50, 64].

In this paper, we are interested in how the benefits of cooperative learning can be made available on-demand to remote learners of 3D-design software, both to address some of the above challenges with tutorials, and to extend the benefits of over-the-shoulder learning to users who would not be able to benefit from it otherwise (e.g., users who are learning at home, either informally or through online courses). To this end, we developed *TwoTorials* (Figure 1) a tutorial system that allows pairs of remote users to complete a tutorial in parallel, with mechanisms to facilitate beneficial learning interactions. Through the design and development of TwoTorials, and two studies, our work addresses the following research questions: (1) *What are the salient components of cooperative learning for step-by-step 3D-design tutorials, including potential benefits, challenges, and common strategies*? and (2) *What are the appropriate design principles and interface features to support these components*?

To address these questions, we first conducted an observational study of four pairs completing step-by-step tutorials in Tinkercad, a popular 3D solid modeling application. The results of this study revealed potential benefits, challenges, and coordination strategies between users cooperatively completing step-by-step tutorials, such as the need to rapidly establish shared context to support their communication, and a hesitance to help one another if not explicitly asked.

Based on this initial study, we derived a set of five design principles for cooperative software tutorial systems and instantiated these in our TwoTorials prototype. The system provides mechanisms to help establish shared context, synchronize user progress, and facilitate non-disruptive communication between the peers.

To evaluate TwoTorials, we ran a second user study with six pairs of participants, comparing a baseline system with minimal coordination features to the TwoTorials system. Our findings, based on a within-subjects mixed-methods user study, indicate that TwoTorials helped participants to complete tutorials faster, significantly reduced their effort and mental demand, and helped them to maintain a higher level of awareness of each other's progress.

Building on previous work in software learnability and cooperative learning, and our interest in fostering peer help for 3D design software tutorials, this work makes three main contributions. First, we contribute a deeper understanding of the potential benefits, challenges, and common behaviors surrounding cooperative learning of 3D design software. Second, based on these findings, we present a set of design principles for remote cooperative tutorial systems, and instantiate these principles in what we believe to be the first cooperative software tutorial system. Finally, a user study contributes an understanding of the benefits of such system for learning feature-rich software, and points to directions for further work, including generalizations to larger peer groups and other software domains.

## 2 RELATED WORK

This work is related to prior research on software learning and tutorial systems, cooperative learning and distributed teamwork, and cooperative interfaces for multiplayer games. We review each of these areas below.

---

[1] *salharth@acm.org*

[2] *ben.lafreniere@gmail.com*

[3] *tovi@dgp.toronto.edu, tovi.grossman@autodesk.com*

[4] *george.fitzmaurice@autodesk.com*

## 2.1 Software Learning and Tutorial Systems

Early research on software learning established a tendency for learners to abandon printed manuals and other learning materials that take time away from their primary task [8, 23, 36]. This has led to a rich body of research on systems and tools to support learning of software applications [13, 25, 78]. In particular, prior work has demonstrated the benefits of step-by-step tutorials and gamified tutorial systems [23, 59], as well as systems that allow users to learn in the context of realistic tasks [28, 35, 78].

A number of research projects have explored methods for harnessing community-created content, or improvements to learning content contributed by other learners, such as improved workflows [11, 37, 58], multimedia demonstrations of tutorial steps [13], or comments on tutorial content [7]. Recent research has also proposed new approaches for how groups learn 3D design [25, 48]. For example, Maestro [25] enables facilitators of 3D modelling workshops to track the progress of their classrooms in real-time, and provides simple mechanisms for provide help to students when needed.

While the above approaches appear to be valuable, research in the education community has demonstrated a range of benefits to active cooperative learning approaches, in which learners are able to directly interact with one another [14, 15, 52] (discussed in detail in the next section). To provide such an active learning experience, some work on software tutorial systems has integrated elements from games [21, 57, 59]. For example, CADament [59] enables users to learn 3D design software skills by observing the workflows of opponents in a competitive multiplayer learning game. The system enables competitors to engage in "over the shoulder" learning, but it is not focused on creating an environment where learners can work on tasks together to help and benefit from each other. Currently, there exists no step-by-step tutorials that explicitly support remote cooperation. To fill this gap, we build on this body of prior work but focus on supporting active cooperative approaches for learning 3D design software with other users and propose the first known step-by-step tutorial system specifically designed to support remote peer learning.

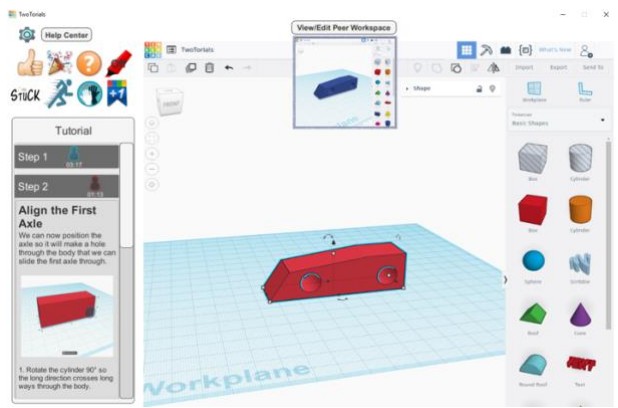

Figure 1: The *TwoTorials* system.

## 2.2 Remote and Cooperative Learning

Remote learning is becoming increasingly prevalent in our world. The global coronavirus (COVID-19)[1] pandemic has demonstrated the need to establish effective remote learning environments [3, 10, 70], and is forcing educators and learners to rethink

---

[1] The 2019/2020 global novel coronavirus (COVID-19) pandemic [10]

pedagogical methods and approaches [6, 20, 81]. A key question raised in this context is how to enable remote learning that preserves the social aspects of in-person learning, giving learners opportunities to engage and interact with each other, which has been shown to aid in motivation and creating positive learning experiences [16, 31, 47, 76, 91]. In the present work, we aim to support remote synchronous coordinated learning for the specific learning resource of step-by-step tutorials.

Collaborative learning is an educational approach in which learners work together to solve a problem or complete a task, recognizing that learning is a naturally social activity [22, 65]. There are many approaches to foster collaborative learning, and significant literature showing its effectiveness both in co-located and remote learning environments [22, 44, 72]. Cooperative learning is a particular type of collaborative learning in which a set of processes help people interact together to accomplish specific goals, helping themselves and others to learn [64, 65, 85]. In terms of specific benefits, social activities have been shown to increase the motivation to learn from others, and to result in effective knowledge gain and transfer [16, 18, 75]. Critically, this kind of social learning does not need to consist of continuous interaction between the learners [64, 73] – simply being able to work together in a social environment provides the opportunity for both passive and active learning. For example, "over-the-shoulder learning" can occur from observing another learner while they are completing a task, or by actively engaging in completing the task together [18, 88]. When compared to individual and competitive learning, cooperative learning has been demonstrated to be particularly effective for sustaining learner motivation [63, 71].

In terms of particular mechanisms for enabling remote cooperative learning, prior work has shown that cooperative learning is most effective when learners organize their activities, synchronize their effort, and maintain shared situational awareness [29, 50, 82]. Research on distributed teams has also shown the importance of awareness mechanisms, to enable team members to inform one another of their status [40, 92]. Although team or group awareness can be easily maintained in co-located collaborative environments, it can be difficult in remote collaboration [40]. Thus, groupware research has focused on interfaces and techniques that facilitate communication, increase group awareness, and enable cooperation, such as capturing eye gaze [18, 19] and other awareness cues [12, 92]. A full literature review of groupware research is beyond the scope of this paper but we point readers to existing surveys on the topic [26, 40, 79].

## 2.3 Cooperative Gaming

Games are a prominent example of systems that cultivate cooperative behavior [68, 84, 87]. Cooperative games provide different interfaces and mechanics to facilitate multiplayer interaction [1, 45, 55]. In cooperative games, mutual understanding of the objectives between the players are essential to their success. Players must maintain awareness of each other and establish a common ground for communication [9]. Cooperative games also provide players with a variety of explicit and implicit communication mechanics, including awareness cues and cooperative communication mechanics [2, 12, 67, 87, 92]. These mechanics help teams communicate with each other and maintain high level of awareness. In this work, we draw on the body of past work on multiplayer games and gamification to develop interfaces to foster effective cooperative learning for a qualitatively different domain – step-by-step tutorials for 3D design software.

In summary, our current work contributes to the understanding of cooperative learning for the domain of step-by-step tutorials for 3D design software, and the TwoTorials prototype system

provides specific mechanisms to enable shared awareness and support cooperative learning in this domain. To the best of our knowledge this work represents the first application of a cooperative active learning approach to step-by-step software tutorial systems.

## 3 OBSERVATIONAL LAB STUDY

To inform the design of our cooperative step-by-step tutorial system, we conducted an observational study with pairs of participants. Our main goal was to understand how peers cooperate with each other to complete this type of tutorial, the challenges they face, how they synchronize their work, and how they encourage and support each other.

### 3.1 Study Procedure

Each pair of participants completed two step-by-step tutorials for Tinkercad drawn from those provided in-product in the software (Balloon Powered Car and Roman Dome), each lasting ~30 minutes, followed by an individual survey and a short semi-structured interview.

We intentionally tested a range of different cooperative tutorial setups (Figure 2), to gain a broad set of insights into the benefits and challenges that arise in different forms of collaboration. These included co-located vs. distributed setups (simulated by a partition between the participants, which permitted them to talk to each other, but still required the use of screen sharing to view each other's workspaces), and separate vs. shared workspace setups (i.e., whether both participants were working on one project together, or working on the same project in parallel). Our decision to use a partition to simulate a distributed condition was designed to reduce the complexity of the study setup, and is an approach that has been used in prior work [18]. In this study, each pair of participants completed two tutorials across one of the axes of the four setups shown in Figure 2, enabling them to comment in greater detail on the effect of that axis.

The experimenter took observations and provided participants with assistance with technical difficulties but did not help the participants with completing the tutorial instructions. Video and audio recording of the study and post-study interviews were transcribed and analyzed for common themes. Each study session lasted ~60 minutes total.

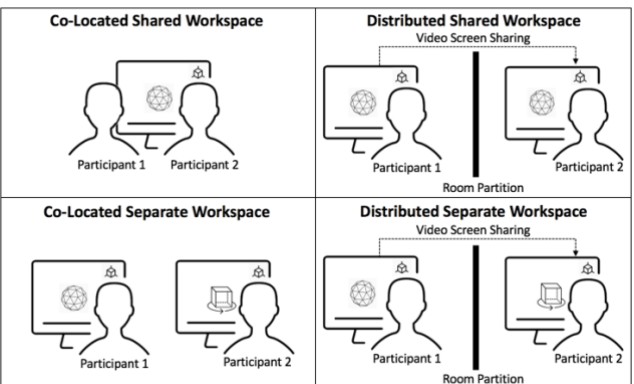

Figure 2: The four cooperative tutorial setups that were tested.

### 3.2 Analysis

Interview transcripts and observations were analyzed using methods drawn from grounded theory [33]. Specifically, open coding was used to label transcript data, and emerging themes and patterns were identified by the first author and then shared and discussed with the broader research team. The themes that emerged relate to the potential benefits to users from cooperative learning of 3D design software, challenges experienced by peers when learning 3D design software together, and common strategies used to cooperatively learn.

### 3.3 Participants

Four pairs (8 participants total (6 male, 2 female), mean age 38.4 years (SD 10.3)) was recruited via an email to employees of a large software company. As pairs volunteered together, they are best considered as coworkers or friends. Two pairs were all male, and two were mixed (one male, one female). All participants reported having completed a bachelor's degree. All participants were screened for prior experience with 3D modeling software; 1/8 participants had no experience, 3/8 participants had minimal experience, 3/8 participants had some experience, and 1/8 participants had extensive experience. The most common 3D modeling applications used previously by participants were Maya, Blender, and Alias. Only one participant had prior experience with TinkerCAD. Each participant received a $25 gift card as compensation for their participation.

### 3.4 Results

We begin by discussing our observations of the tradeoffs of separate vs. shared workspaces and co-located vs. distributed workspaces, and then discuss our findings on the benefits, challenges, and strategies used by participants to cooperatively complete step-by-step tutorials.

#### 3.4.1 Separate vs. Shared Workspace

Neither the separate nor shared workspace setups were revealed to be clearly superior for enabling cooperative learning, with both showing advantages and disadvantages. Having a shared workspace forced the peers to collaborate, which was beneficial, but created a situation where the participant that was not 'driving' the system could become frustrated, and feel like they were missing out on learning:

> When I was just watching it was frustrating to not be able to take actions myself. We were trying to figure out how the interface works, and I want to be able to create my own objects to explore the manipulators and what is possible. (P5)

Conversely, participants reported that working on separate workspaces created a feeling of working in parallel on separate tasks:

> We had the video sharing, but we were both doing our own thing, so we only looked at each other's views to make sure our work looked somewhat similar. It didn't really seem like a cooperative effort [in the distributed separate workspace condition], more like we were just doing the same thing at the same time. (P3)

This observation is consistent with prior work on personalizable groupware that can support individual and group activities [34, 55]. A design approach that emerged from this observation was the idea of a hybrid system, which would allow each of the peers to benefit from actively working on the tutorial individually, while encouraging cooperation and peer help.

#### 3.4.2 Co-located vs. Distributed

Contrasting the co-located and distributed setups revealed a range of challenges to coordinating effort when participants were

distributed. When co-located, it was much easier for participants to make spatial references to parts of the 3D environment and to assist one another by looking at each other's screens, pointing at parts of their peer's screen, or even taking over the mouse of their peer to rotate the camera or make simple changes to 3D objects. Consistent with prior work on collaborative remote physical tasks [29, 54], cooperative help-giving and receiving in a 3D design tutorial was much more difficult when participants were distributed. Participants were not always able to clearly understand each other due to a mismatch in their respective views of the 3D environment, and providing verbal instructions became complex without the ability to ground the instructions in spatial references, or to make direct changes:

> Explaining how I want my partner to try using the manipulator with words is much slower than just being able to do it myself. (P4)

These observations suggest that additional coordination mechanisms are needed to enable productive cooperative learning in distributed setups.

### 3.4.3 Benefits of Cooperative Learning of 3D Design Software

In terms of the benefits to cooperative learning of 3D design software, participants reported having an overall positive experience, and suggested that this approach allowed them to gain additional insights beyond the tutorial content:

> When we both were doing the tutorial, it just felt that, wow, that moved on very quickly, and [I] actually still learned something, and it might not [have] been what was intended to be learned through the steps, but, like, the other person's insights. (P2)

Participants also pointed out the benefit of being able to quickly detect errors and identify if they were misunderstanding the tutorial instructions:

> I think working together has a lot of advantages. You can detect errors very quickly and keep on making progress. (P5)

Participants also indicated that cooperatively working on a tutorial helped them to accelerate their learning and sustain motivation to learn the tutorial content:

> It accelerated the learning since it was a shared experience, and we could communicate what our successes and failures were to each other. (P8)

Overall, these observations point to several potential benefits of cooperative learning of 3D design software, which are worthy of further investigation.

### 3.4.4 Strategies for Cooperation

Our observations and interviews indicated several common strategies that peers used to cooperate with one another. During help-seeking instances, we observed that participants in distributed setups started by establishing common ground and shared 3D context with their peer as a first step when providing assistance. For example, they would ask questions such as, "which view are you on – top, side, bottom?" And then they would change to that view and proceed to make recommendations and provide help:

> Got to first understand the language and perspective and then give feedback after. (P4)

This strategy was more common when participants were distributed. Related to this theme, we observed that peers would frequently communicate which step in the tutorial they were on, or signal to their peer that they are moving on to the next step, as suggested by the following quote in response to a question on what techniques or practices P4 was using to work together and synchronize their efforts with their peer:

> Make sure to communicate that we were on the same step and sub-step. (P4)

This strategy suggests that mechanisms for establishing shared context between learners could be beneficial, particularly if they can support the sharing of 3D viewpoints and the step of the tutorial a learner is currently on.

A final beneficial strategy we observed was that looking at their peer's workspace provided learners with insights into their own work and how it could be improved. This over-the-shoulder learning [88] was observed in multiple instances where peers would spend time observing each other completing a step of the tutorial and then attempt it themselves.

> I can look at the other person's work and say, mine doesn't look anything like that, and then you know there is a problem. (P1)

In terms of design guidance, providing explicit support for this learning strategy could be valuable.

### 3.4.5 Challenges in Cooperative Learning of 3D Design Software

Finally, we observed several challenges that can come from working cooperatively on step-by-step tutorials for 3D design software. For example, participants struggled to maintain awareness of each other's activities, due to a lack of shared context:

> Sometimes it is difficult when you can not see what I'm seeing, you would be like, oh it is like this, and I'm like no it is not, we both are seeing different things, and we are arguing about nothing, that was frustrating. (P5)

While maintaining awareness is a known issue for groupware applications [27, 40–42, 92], these challenges appeared to be exacerbated from working in a 3D environment, where each peer had a different camera orientation on the scene, making it difficult to establish shared context when help is needed:

> At some times, we both were oriented in different ways, and I'm like something is wrong here, and she is like, oh no we just need to adjust the orientations to match. (P1)

Another challenge reported by participants was that it was difficult to determine when feedback or help was needed or would be welcomed by their peer. This was especially prominent in the distributed setups, where awareness of activities between peers was less strong. We believe that the domain of 3D design software exacerbates this problem, because the editing history of a model, or any mistakes made on previous steps, are not obvious to a peer observing the model "over the shoulder" as the user works on it.

Finally, while not necessarily a challenge specific to 3D design software, peers found it difficult to synchronize their progress in the step-by-step tutorials. In the distributed setups, participants

would verbally share which step they were on in the tutorial, to help each other maintain constant awareness of each other's progress, and to signal if one of them was falling behind and might need help. This was less of a problem in the co-located setups, where participants could glance at their peer's screen to get the same information:

> I found that my partner jumped to the next step. I needed to confirm what step he was on so that I could help with the next step (P4)

Past work has suggested that this kind of communication overhead can be distracting [62], which can detract from the learning process. Thus, it may be beneficial to design features to reduce this "orienting communication" overhead.

## 4 DESIGN PRINCIPLES

The results of our observational study complement prior literature and provide an understanding of the main challenges and breakdowns faced by peers learning 3D design software through step-by-step tutorials. Our findings are consistent with known issues surrounding control ownership, and collocated use of groupware systems, but also reveal important and unique insights specifically related to both cooperative use of step-by-step tutorials and learning challenges for 3D software. Pulling together the observations and insights from the study, we suggest a set of five design principles for cooperative step-by-step tutorial systems for 3D design software:

### 4.1 Help Establish Shared 3D Context (D1)

The system should assist with establishing and maintaining shared 3D context between peers, to make giving and receiving help easier. The need to establish and maintain shared 3D context has been identified in prior work [24, 30, 90], but it presents a particular challenge for step-by-step tutorials, where each user may have a different camera position and orientation on a separate 3D workspace, whose 3D content may be at a different stage of the tutorial than their peer. This makes it difficult to establish the context necessary to reference 3D objects or meaningfully discuss their orientation.

### 4.2 Balance Independent Action with Encouraging Collaboration (D2)

Two competing challenges we observed were peers becoming frustrated with not being able to 'drive' in the shared workspace condition, and peers not engaging with each other in the separate workspace condition. Prior work has shown that giving users the power over navigation, manipulation, and representation within shared workspaces supports collaboration, but has its tradeoffs [39]. The system should balance the need for independence, while also encouraging collaboration between the peers, to create a beneficial cooperative learning experience where both users are engaged with the tutorial task.

### 4.3 High-Level Awareness of Progress (D3)

The system should provide learners with high-level awareness of where their peer is in the tutorial steps and in the 3D workspace and help make learners aware of any challenges or setbacks faced by their peer. This is particularly relevant for 3D environments, where it is more difficult to maintain awareness and establish mutual orientation and view between peers [24, 30].

### 4.4 Non-disruptive Communication Mechanisms (D4)

The system should provide non-disruptive communication modalities that simplify and complement the beneficial cooperative learning practices that we observed. Prior work suggests that communication can be less disruptive when timing and communication method are selected appropriately [17].

### 4.5 Synchronize Progress (D5)

To increase the likelihood of beneficial cooperation, and to try and avoid the situation where one learner quickly finishes the tutorial and becomes bored, the system should encourage peers to work together and synchronize their progress through the tutorial steps.

Guided by the principles above and previous research in this area, we developed TwoTorials, a cooperative step-by-step tutorial system designed for pairs of learners, which we present next.

## 5 THE TWOTORIALS SYSTEM

TwoTorials offers a cooperative learning environment for two distributed users. The pair of users work cooperatively to learn 3D design software by each completing the same step-by-step tutorial in parallel (Figure 1). The system includes a set of features to support coordination and establish shared context within the tutorial. In the current work, we designed the system to work with Tinkercad, a popular web-based 3D solid modeling tool [4]. In this section, we start with a high-level overview description of TwoTorials, then we highlight the main features, noting in parentheses the relevant design principles each of these features is intended to address, and citing any prior work that influenced the designed features.

### 5.1 System Overview

Each remote peer gets an individual workspace, as well as access to a constantly updating and editable view of their peer's workspace, enabling users to observe their peers and actively assist them if needed. Peers can communicate, help, and encourage each other using both verbal and non-verbal communication modalities. The system also provides implicit awareness cues to helps learners maintain awareness of their own progress in the tutorial and whether their peers are falling behind and may need help; and can enforce a level of interdependence between the learners to encourage them to work together and help each other out. Finally, as a user completes each step, their peer is provided with a screen recording of their efforts on that step, providing further material for the peers to reference when helping one another. The sections that follow describe the above features in detail.

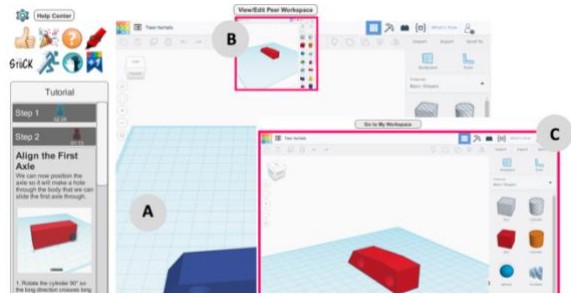

Figure 3: User and peer workspaces. (A) the user's workspace; (B) a constantly updating view of the peer's screen; (C) an expanded view of the peer's screen (accessible by clicking the small view).

## 5.2 Seamless View, Transition, and Editing between Workspaces

In TwoTorials, each user gets a small, constantly-updating view of their peer's screen (Figure 3B) displayed above their own workspace (Figure 3A). Clicking the small view of their peer's screen expands it to full screen (Figure 3C) and allows the user to directly edit their peer's workspace. Through these mechanisms, learners are able to constantly monitor their peer's progress, enabling over-the-shoulder learning [88], and helping to establish shared context (D1). The ability to make changes to their peer's workspace in the expanded view enables a user to directly provide assistance or demonstrate editing operations on the peer's 3D model (D2). Prior work has shown these kinds of seamless transition mechanisms from individual to shared spaces to be important for facilitating collaboration [29, 32, 53, 83].

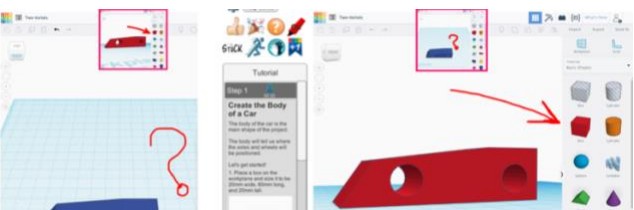

Figure 4: Drawing Annotations enable the user to draw over the 3D workspace, enabling cooperation and conversational grounding between peers [2, 44].

## 5.3 Verbal and Non-Verbal Communication Features

The system provides in-tutorial voice and text chat, allowing peers to verbally communicate or send text messages to each other (D4). To complement these communication methods, the user can also create free-hand drawing annotations on top of both workspaces in the form of free-hand lines and simple shapes (Figure 4) [2, 74]. These non-verbal communication help users to ground their conversations or direct their peer's attention to parts of the UI or 3D workspace (D1). Prior work has shown such mechanisms to be effective for keeping users engaged in collaborative activities in games [2, 89] and online courses [44].

Users can also send peer pings, a set of predefined visual messages that provide simple, non-disruptive communication between users. Clicking on one of these pings (Figure 5), sends a visual message to their peers that lasts for a couple of seconds, displayed on top of their peer's workspace. Each of these pings is designed to indicate a situation where cooperation is needed, such as having a question, being stuck, or expressing the need to move faster (D5). Peer pings are also supported to celebrate success or provide encouragement, such as sending fireworks, high-fives, and thumbs ups (D2). This type of lightweight communication mechanism has been shown to be effective in encouraging participation in gaming and live streaming contexts [43, 44, 69].

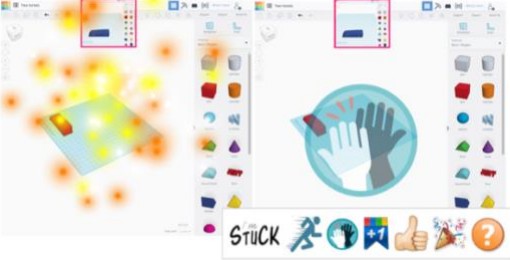

Figure 5: *Peer Pings* are predefined visual messages that provide lightweight communication between peers.

## 5.4 Implicit Awareness Cues

To enable users to maintain awareness of their peer's progress (D3), each user has a simple avatar that moves through the list of tutorial steps as they proceed though the tutorial content (Figure 6). A timer indicates the amount of time the user has spent at the current step, further fostering peer awareness. Finally, to provide spatial awareness [41, 42] of a peer's activities within a step, the system displays the peer's mouse cursor in the user's workspace.

The non-verbal awareness cues described above allow users to maintain awareness of each other's activities in a lightweight manner, without the need to constantly communicate their status explicitly (D5). Similar mechanisms have been shown to be important for enabling users to maintain shared awareness [40, 92], especially for distributed environments, which lack the sensory cues that ease collaboration in co-located settings [12].

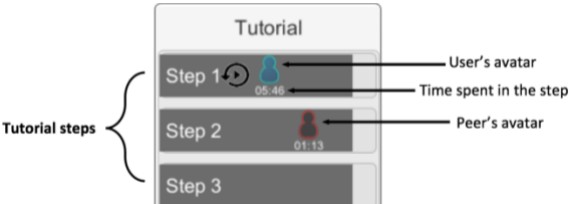

Figure 6: Tutorial-progress awareness cues, including the user avatar, and indicator of time spent on the current step.

## 5.5 Progress Control Mechanism

Before starting a tutorial, users can select one of three levels of step-synchronization, which affect how much the system enforces synchronization of activities between the peers (Figure 7). At the most extreme, the Strict setting prevents each user from moving on to the next step until they are both finished the current step (indicated by clicking a button). The Moderate setting enables a user to move one step ahead of their peer, and if they try to move any further, they are prompted to wait. Finally, the Free setting puts no restrictions on movement through the steps. These mechanisms provide system-imposed synchronization of progress (D5), primarily motivated by our observational study results.

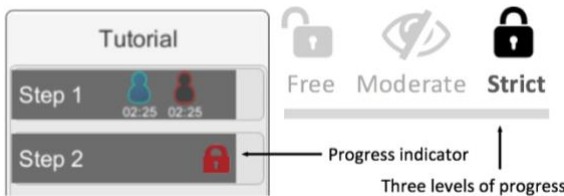

Figure 7: The progress control mechanism enables users to control the progress of each peer in the tutorial.

## 5.6 Workflow Replay

The system records a video of each user's screen as they work on a step and makes this recording available to their peer upon proceeding to the next step. By clicking a replay icon, the peer can view a video showing the exact steps the user took to complete that step (Figure 7). Past work has shown that this kind of short demonstration video can be particularly valuable for learning design software [37, 59], and this feature also frees a user from having to explain the exact process they followed – they can simply prompt their peer to check the recording video (D2).

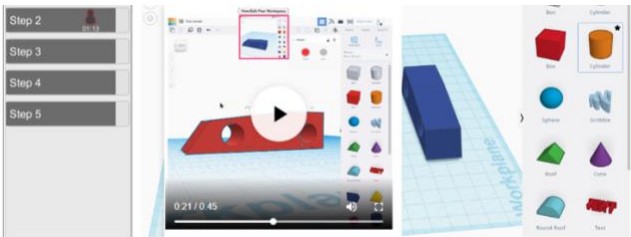

Figure 8: Video replay window.

## 5.7 Access to Online Help Resources

The system provides quick in-application access to online and community-based help resources (e.g., the Tinkercad help center). This enables users to access help without disengaging from the tutorial experience (D2).

## 5.8 System Implementation

TwoTorials was implemented in two parts. First, the step-by-step tutorial system was built as a Unity application. This enabled us to quickly build a multi-user system by taking advantage of Unity's networking capabilities to provide a reliable, low-latency connection between the peers for sending media streams, including voice and text chat, user progress data, shared annotations, and peer pings. Screen recording and playback was implemented using a Unity plugin that enables real-time video and audio capture and streaming. The Tinkercad application was embedded using a Unity web-browser component, which mirrored a locally running version of Tinkercad. The second part of the system consisted of a modified version of the Tinkercad application to add required concurrent editing features to the application.

## 5.9 Tutorial Format, Authoring, and Progress Tracking

Each tutorial step consists of text and images (Figure 1, left). We adopted this format to match as close as possible the in-product tutorials available in Tinkercad, which we used for the baseline condition in our evaluation study, explained in the next section. In terms of tutorial authoring, text was manually entered, and figures were added to a folder that was read by the system. TwoTorials tracks progress solely based on navigation through the tutorial steps (users explicitly clicking "next step"). More sophisticated tracking of Tinkercad tool usage or the 3D content being created is an interesting avenue for future work.

## 6 EVALUATION

We conducted a user study to understand users' reactions to the TwoTorials system and its cooperative features, and to gain further insights into the cooperative experience of step-by-step tutorials.

## 6.1 Study Procedure and Design

The study followed a within-subjects mixed-methods design, with each pair of participants completing two step-by-step tutorials, one using TwoTorials, and the other using Tinkercad's built-in tutorial interface. These tutorials were the same as those used in the previous observational study, which had revealed them to be about the same level of difficulty. For the TwoTorials condition, the progress control setting was set to 'Free'. Although in-application voice chat was implemented in the system, the setup of the study resulted in us not needing to use it – participants were simply instructed to talk with each other over the divider (similar to our observational study and methods used in prior work [18]).

In the baseline condition, participants used the Tinkercad tutorial along with a live screencast of their workspace, shared with their peer through Google Hangouts. We provided this capability in the baseline condition because it seemed unrealistic for users to collaborate with no view of their peer's workspace whatsoever. Participants in this condition were also able to talk with each other over the divider.

To rule out ordering and learning effects, condition order and mapping of tutorials to conditions was fully counterbalanced.

At the start of the study, participants were provided informed consent, and asked to complete a questionnaire on demographics and prior 3D design software experience. Next, the experimenter introduced the study system, and the available cooperative features, before allowing the participants to work on the tutorial. The experimenter did not help participants with working through the tutorial instructions but did provide limited assistance in response to technical difficulties with the study system. After completing each condition, a set of Likert-style questions were administered on the overall experience, ease of following the tutorial, learning, and usefulness of the cooperative features. The NASA-TLX questionnaire was also administered, to assess workload [45, 46]. At the end of the study session, a post-study open-ended questionnaire was administered. The study took ~60 minutes total to complete.

## 6.2 Participants

Six pairs (12 participants total (10 male, 2 female), mean age 35.8 years (SD 7.9)) was recruited via an email to employees of a large software company. Each pair was either friends or co-workers, with 1 pair all female, and 5 all males. All participants were screened for prior experience with 3D modeling software; 1/12 participants had no experience, 4/12 participants had minimal experience, 5/12 participants had some experience, and 2/12 participants had extensive experience. Most common 3D modeling applications used previously by participants were Fusion 360, Maya, and SolidWorks. Only two participants had prior experience with TinkerCAD. Each participant received a $25 gift card as compensation for their participation.

## 6.3 Results

We begin by presenting the main quantitative findings, comparing TwoTorials to the baseline. We then present results from the post-condition questionnaire, and the usage and subjective ratings for TwoTorials features. Finally, we discuss our qualitative and semi-structured interview findings.

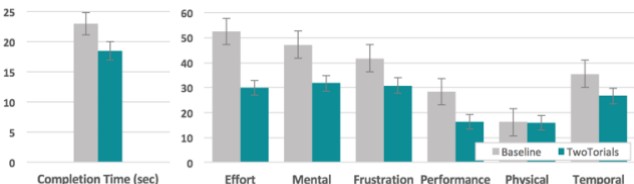

Figure 9: Completion time in seconds and NASA-TLX results (lower is better).

### 6.3.1 Performance Results – TwoTorials vs. Baseline

A Wilcoxon Signed-Rank Test showed that pairs spent significantly less time to complete the tutorial together using TwoTorials (M = 18.5) compared to the Baseline (M = 23) (z = 2.831, p<.05) (Figure 9). These findings provide evidence that the features of TwoTorials helped participants to complete the tutorial together more quickly.

### 6.3.2 Cognitive Load Results – TwoTorials vs. Baseline

For the cognitive load results (Figure 9), a Wilcoxon Signed-Rank Test showed significantly lower rating for effort (z = 2.668, p<.01), mental demand (z = 2.201, p<.05), and frustration (z = 2.254, p<.05) for the TwoTorials condition as compared to the baseline condition. These findings provide compelling evidence that the features of TwoTorials helped reduce the cognitive load on participants. For the rest of the TLX subscales, we found no significant difference.

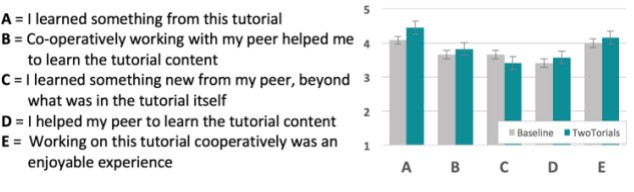

Figure 10: Rating on the learning experience questionnaire (higher is better). Error bars show standard error.

### 6.3.3 Questionnaire Results – TwoTorials vs. Baseline

When asked which of the two conditions they preferred overall, TwoTorials was rated higher by 5/12 participants compared to the in-application tutorial, with 6/12 participants expressing no preference and 1/12 preferring the baseline condition. While this suggests a preference for the TwoTorials system, a Wilcoxon signed-rank test did not show this difference to be statistically significant.

For each condition, we asked participants a set of questions on what they learned from the tutorial experience (Figure 10). For most of the questions we found no significant difference, but a Wilcoxon Signed-Rank Test showed a significant difference in medians for the statement "I learned something from this tutorial" favoring the TwoTorials condition over the baseline condition (z = 2.000, p < .05).

We also asked participants a set of questions on the various other aspects of the tutorial-following experience (Figure 11). A Wilcoxon Signed-Rank Test determined that there was a significantly higher median for the TwoTorials system for "maintaining awareness of your peer's activities" as compared to the baseline (z = 2.197, p < .05). We did not find a significant difference for the other questions in this group.

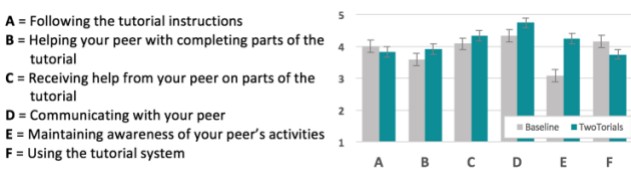

Figure 11: Ratings of the tutorial systems for various statements. Error bars show standard error.

### 6.3.4 TwoTorials Features

For the TwoTorials condition, we analyzed how many times each feature was used by participants, and asked participants to rate the usefulness of the individual features. In terms of usage, participants switched to their peer's workspace an average of 4.8 times (SD=1.94) and edited their peer's workspace directly 2.2 times (SD=0.75). Participants annotated each other's workspaces 2.3 times (SD=1.03) and sent 4.6 peer pings (SD=2.42). Considering that pairs in the TwoTorials condition took less than 25 minutes to complete the tutorial, these numbers suggest that the features of TwoTorials were used frequently by participants.

The ratings of usefulness for the individual features of TwoTorials are shown in Figure 12. Participants generally reported the features to be useful. There was strong support for the voice chat, the ability to view the peer's workspace, and the ability to directly edit the peer's workspace. The only feature to receive a strong negative rating for usefulness was the text chat, which is likely because the voice chat provided a much richer and more convenient communication medium.

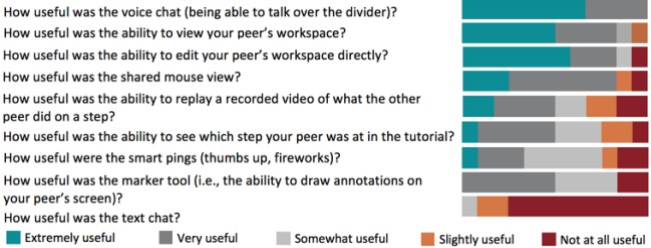

Figure 12: Rating of individual TwoTorials features.

## 6.4 Participant Feedback and Observations

At the end of the study session, we asked participants to contrast the experience of working with TwoTorials and the baseline tutorial system. Qualitative data were analyzed using methods drawn from grounded theory [33]. Specifically, open coding was used to label the data and emerging themes were identified by the first author and then shared and discussed with the broader research team.

### 6.4.1 Improved Communication, Awareness, and Coordination

Participants reported being able to coordinate with each other more effectively using the TwoTorials system, with smoother information flow between peers. Participants noted that having a constant view into their peer's workspace helped them solve problems more effectively without breaking the flow of working on the tutorial:

> Having the constant visual of my peer helped quite a bit to solve common problems on my workflow instead of having to stop the flow to find the assistance. (P11)

Participants also appreciated the ease with which they could switch from viewing their own workspace to that of their peer:

> The live view of your companion was a big plus. Easily being able to switch to their view and affect their workspace is a big plus as well. (P12)

Participants reported that TwoTorials helped them to maintain an ongoing awareness of the other user, and this helped to encourage dialog:

> The first system [TwoTorials] reminded me to think about discussing, because the view of the other screen was always present […] it helped slightly by encouraging dialog. (P9)

Participants also described using the shared awareness features to ground their discussions with their peer:

> It helped to see where the person was so we could say "look at my screen this is what you're supposed to have." (P6)

### 6.4.2 A Cooperative Learning Environment

A second common theme was that the TwoTorials features created an environment where cooperative learning was supported. Along these lines, one feature cited by participants was the ability to directly edit their peer's workspace. We observed several occasions where one peer would provide help by directly making changes in the workspace of their peer. Participants reported that this was an efficient way to help each other:

> The fact that I could work directly on my peer's workspace in [TwoTorials], let me help him more efficiently. (P7)

Participants also expressed appreciation for the annotation features, and highlighted how it created more of a "lesson experience" than a tutorial:

> In [TwoTorials], the fact that my peer could chime in and add his notes in real time made it more of a lesson experience than a tutorial - the chance to clarify and question each other as we followed the steps was a very useful addition. (P11)

This quote is particularly encouraging because it suggests the features of the TwoTorials system were able to change the experience to one where cooperation and helping each other was more natural. Along similar lines, P8 suggested that TwoTorials could be used in formal educational settings to enable teacher-student interactions:

> In [TwoTorials], getting help was much easier. I would imagine a TA or teacher helping students through that system. (P8)

Participants also commented that they took advantage of the expertise of their peer less in the baseline condition:

> If I got stuck, the person knew exactly where I was (they were there too or had just been there) and most likely had the same problems. I used the person less [in the baseline condition]. (P6)

Overall, this feedback provides validation that the TwoTorials features encouraged cooperation and helped to create an environment that supports cooperative learning.

### 6.4.3 Motivating and Enjoyable Experience

Finally, participants reported enjoying the cooperative tutorial experience (in both conditions), and found it to be engaging:

> Working cooperatively was fun and kept me engaged. Also, I learned some tips from the other person. (P9)

While participants reported enjoying the experience of cooperating in both tutorials, some participants noted that TwoTorials enhanced this aspect of the experience:

> In [TwoTorials], the second layer of interaction added a different [kind] of enjoyment, where we could interact and made the experience more fun. (P11)

A specific feature cited as creating an enjoyable experience was the peer pings. Four participants stated that they felt the peer pings were fun and helped encourage them to cooperate:

> "chat icons" were a nice touch to encourage each other. (P11)

> There was a sense of competition that reduced co-operative work in both tutorials. This was less so in [TwoTorials] because of the added features like thumbs up etc. (P8)

This final quote is particularly encouraging, it suggests that peer pings were able to reduce the sense of competition between the peers, which could stand in the way of the cooperative experience the system is designed to foster.

## 6.5 Challenges Encountered

While participants were generally supportive of the features of TwoTorials, some features elicited mixed feelings. Specifically, the ability to directly modify content in a peer's workspace was cited as undesirable by some participants:

> I do not want to interfere with my partner's screen. Annotation can be helpful though, and stickers [peer pings] make it more fun, but not direct interaction. (P1)

> I did not feel comfortable editing my partner's workspace. (P9)

As we discuss in the next section, we believe this indicates the need for better social mechanisms to be built around these features, to ensure that they can only be used to provide help or edit a peer's workspace when that help is welcome, as suggested by prior work on collaboration boundaries [83].

More broadly than any individual feature, one of the participants expressed that he would prefer to work on his own, because he did not like being observed while he worked:

> I personally like working on a tutorial alone and having others watching my work is kind of irritating. (P8)

This is important feedback, but in practice we believe that those who are interested in cooperative learning will choose to use TwoTorials or other systems like it, while those who are not can continue to use the many resources currently available to support individual learning.

## 7 DISCUSSION AND FUTURE WORK

Overall, our evaluation indicated that TwoTorials helped participants to engage in cooperative learning, improved their performance, reduced effort, and mental demand, and helped participants to maintain awareness of each other's progress in the tutorial. Feedback from participants also suggests that the system's features helped to create a supportive environment for cooperative learning, helped keep learner motivation high, and helped foster a feeling of cooperation rather than competition between the learners. These are promising findings for applying the cooperative software learning approach to step-by-step 3D design software tutorials.

While our study results are generally encouraging, we found that some participants did not appreciate the ability to directly allow peers to edit one another's workspaces. This is important feedback, particularly because this study was conducted with peers who knew each other as friends or colleagues – it seems likely that learners will be more hesitant about this feature if they were working with peers with whom they don't have an existing relationship. To overcome this challenge, we believe that simple

permission mechanisms could be put in place. For example, a user could be prevented from editing their peer's workspace unless that peer explicitly asks for help and provides editing permission. Editing permission could also be limited to a short period of time, or to a selected subset of objects in the workspace. This approach would fit with prior research on groupware and MOOCS, which suggests that each user should have their own territory [44], with permission and roles mechanisms to enable users to control who can view and edit [77, 83]. Alternately, the system could enable a "forked demonstrations" paradigm, where a user could get a copy of their peer's current workspace that they could edit to demonstrate an operation to their peer, without making any lasting change to the peer's workspace itself.

## 7.1 Matching with Remote Peers

In this paper we focused on investigating features that could enable a cooperative learning experience for distributed pairs of users working on step-by-step software tutorials. Having established the benefits of this approach, a next important question is how to match pairs of remote users to work together on tutorials. There are several interesting possibilities here. The results of our observational study suggest that it may not be a good idea to match users with large differences in overall experience and expertise, which could result in the more experienced user becoming bored. Instead, the system could try to match users who are at similar levels of experience but have complementary skill sets. It would be particularly interesting if the system could consider both the skills of the learners and the required skills for the tutorial, to create an experience where peers would need to work together and help one another to reach the goal. These skill-based matchmaking mechanics could be designed in a similar way to those available in multiplayer games [1, 66].

## 7.2 Additional Peers

Another interesting area for future work would be to consider how the cooperative software tutorial approach could accommodate more than two learners. An advantage of the approach we have adopted, where each user is working in parallel on the tutorials, is that it could naturally support additional peers – in contrast, if more than two people were working on one shared workspace, it could quickly become unwieldy. The advantage of adding additional peers is more collective expertise, which could help get the group of peers unstuck when they face challenges. However, this could also create additional conflicts between users, or situations where certain users pair off, leaving others out. These challenges make this an interesting area for investigation, and we see the potential for a scaled up system to be used as a component of interactive 3D design MOOCS [38, 44].

## 7.3 Beyond 3D Design Software

Although we focused on step-by-step tutorials for 3D design software, we believe that the features of TwoTorials could be easily adapted to work in other software domains with a strong visual element, such as photo editing or the creation of games using game engines (e.g., Unity). From a technical standpoint, our system could be used with minimal modifications with any web-based software application.

## 7.4 Limitations

This work adds to a growing body of research on software learning (e.g., [13, 36, 51, 59, 78]) and provides insights into how step-by-step tutorial systems can be adapted to support remote cooperative learning. However, there are several limitations to this work which should be addressed in future research. First, our study was conducted with a small, specific sample (employees of a software company), which may limit the generalizability of the findings. Many of the findings in our study were not statistically significant and one potential reason is the small sample size. A good next step would be to increase the sample size and deploy TwoTorials in a large online 3D design course, with remote students. Second, participants in this study had an already established relationship before starting the tutorials as either friends or co-workers, which positively helped them collaborate more effectively. Future work should take into consideration the differences in relationships between learners and how it might affect their collaboration and learning. Third, TwoTorials was compared against a baseline that offered minimal coordination features. This was intentional, in order to reveal which of TwoTorials' features were most useful to support collaboration, but future work should compare these features to those offered in state-of-the-art online collaborative learning solutions, such as free-form web curation tools [44, 61]. Forth, prior research has shown that ethnocultural norms and backgrounds can influence the effectiveness of cooperative learning [49, 60, 86], so it is important to expand the evaluation of this type of system to a much larger and more diverse set of participants. Finally, we did not collect data on the long-term effects or value of our system in sustaining learner motivation or encouraging more extensive learning of a domain, which would be an interesting avenue for future work.

## 8 CONCLUSION

This work has demonstrated an approach and a set of features for creating cooperative remote software tutorial systems. Our findings indicate that participants enjoy the cooperative learning experience that this approach enables. Overall, we see this work as a first step toward a future where anyone, anywhere can gain the learning benefits of working alongside peers on interesting and engaging projects.

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
