# OpenReview forum: "TwoTorials: A Remote Cooperative Tutorial System for 3D Design Software"
_graphicsinterface.org/Graphics_Interface/2022/Conference — GI 2022_

### Official Review · Reviewer_ZVzS · 2022-04-09
**TwoTorials addresses an important problem in an interesting way and would be a valuable addition to the HCI community.**

**Rating:** 8
**Confidence:** 5

**Review:**

This paper presents TwoTorials, a collaborative learning system built for 3D design software that helps pairs of software users work together on a tutorial. TwoTorials mimics over-the-shoulder learning for remote users by providing an awareness of collaborator activities, social features, and the ability to quickly 'jump' into a collaborator's workspace to help. An observational study informs the design of TwoTorials and another user study evaluates its effectiveness as a tutorial system.

The related work is thoroughly presented and discussed, the system design seems well-informed, and the system evaluation provides valuable insights into the usefulness of such a system. While I do think there are some ways the paper could be improved, overall, I think this paper is of excellent quality and makes an interesting contribution to the HCI community. I would argue for its acceptance.

The biggest thing I was wondering about was the effect of expertise, for both the observational study and the system evaluation. Section 7.1 suggests that collaborator skills should be complemented, but I think this should be discussed more thoroughly in 3.4 and 6.3. For example, I could imagine that those who are less experienced with 3D design would feel even less comfortable taking control of or annotating their collaborator's view. There may also be some interesting interactions between the features used and expertise. For example, maybe those who are more experienced send more encouraging peer pings, especially if they take on more of the "teacher" role (like a TA helping a student). It seems a bit unlikely that all users of varying levels of expertise would make equal use of all features and a detailed breakdown by expertise would really strengthen the paper.

Another broader discussion that would be valuable to include is how TwoTorials could integrate into existing software. Using a cooperative tutorial like this seems very intentional: a user would likely have a clear learning goal in mind in order to engage with a longer and structured tutorial in general, let alone with someone. How could TwoTorials support more ad hoc learning objectives (that may not be tied to a specific tutorial with step-by-step instructions)? It also would be valuable to discuss how it could scale along with expertise (I'm not an expert in this topic specifically, but experts may be less likely to engage in longer, step-by-step tutorials as their needs become more niche and task-driven). The ability to customize the tutorial steps to support a specific workflow may also be beneficial.

A minor issue that can easily be fixed, but it was a little unclear what Figures 10 and 11 were showing. Are the numbers just numerical ratings from 1-5? It was a bit confusing because 6.1 just notes TLX and Likert-style questions in the questionnaire, but it seems like all of the results for the Likert responses are given in Figure 12.

Other:
- 2.3, first paragraph: I think "variety" should be used instead of "verity"
- 3.2: should be "cooperative learning" not "leaning"
- Double check the references: some have first initials for authors, others have full first names (I believe it should be first initials, based on the template's sample PDF)

---

### Official Review · Reviewer_UWT4 · 2022-04-11
**I found the paper very well structured and written and it addresses an important topic of remote learning with collaboration for 3D content. Authors generally did a good job understanding the current challenges and proposing a new system. They further rigorously tested the suggested system with different techniques.**

**Rating:** 7
**Confidence:** 3

**Review:**

Summary:
The study introduces a design remote cooperative software system called TwoTorials that shares 3D content between a pairs of remote users and coordinates between them. The study highlights various benefits of the TowTorials tool including enhanced collaboration between users, reduced effort, and high engagement between learners.

Reasons to accept:
- Well-written extensive literature review of past and current studies
- The supplementary material was really helpful i.e. explaining the system via a video was a great demonstration of the work conducted in the study and a good summary of the paper.
- Different techniques were adopted to evaluate the effectiveness of the TwoTorial system including questionnaire, sumi-structured interviews, etc.

Reasons to reject:
- Authors listed various benefits to cooperative learning. However, this does not always reflect what people experience in this setting. For example, not specifically related to 3D learning, if learners have significantly different skill levels, working together might not be beneficial to both parties equally. Furthermore, there are personality skills that effet the efficiency of cooperative learning and none of these have been mentioned when talking about this learning setting. Those points were briefly addressed in section 7, however more discussion about the limitation of recruitment needs to be added as well and further discussion about individual differences that might effect the success of the TwoTorial system.
- Learners had an already established relationship before starting the tutorials as either friends or co-workers. This type of relationship would positively help the collaboration between learners. It would be intersting to see how results would change in case learners are matched randomly with no prior interaction among each other.
- Many of the findings were not statistically significant and one potential reason is the small sample size. If the sample size included more participants, results might be more helpful and meaningful.

---

### Official Review · Reviewer_xRMc · 2022-04-13
**Overall good work, small stylistic/qualitative issues**

**Rating:** 7
**Confidence:** 3

**Review:**

This paper presents TwoTorials, a concept that enables more fluid collaboration between users through a variety of shared input and output techniques. Beginning with a formative user study, the paper demonstrates the need for a shared, distributed workplace solution that allows for easy collaboration, teaching, and cooperation. After this, the authors describe five design principles for distributed collaborative tutorial systems as well as an overview of the TwoTorials implementation. The authors evaluate the system and show that TwoTorials does help in terms of project completion time, some aspects of cognitive load (via NASA-TLX) and overall self-reported learning experience. The authors end the paper with challenges, limitations, and future work.

The COVID-19 pandemic continues to emphasize the ever-increasing need for remote workers and distributed teams to have a cohesive and seamless cooperation experience. As such, I find the problem that TwoTorials aims to assist especially timely and relevant. TwoTorials provides several elegant solutions, and the use of both qualitative and quantitative data like quotes particularly convincing in demonstrating both the problem space as well as TwoTorials’ efficacy.

I have a few notes from my reading which I’ll list below:
- This is a minor stylistic note, but for ease of reading and simplicity, I would prefer the word “dyad” be swapped for the clearer word “pair”, unless there is a specific reason for the distinction of which I’m not aware.
- In the grounded theory qualitative work, I would have liked to see measures of inter-rater agreement for the coding sections. This would increase the sense of trust that readers have in knowing these values and codes are accurate and reproducible. For example, Cohen's kappa.
- On the graphs, indicating significance with an asterisk or some kind of visual element would aid in readability.

You'll notice that my notes are relatively short, which is a sign that I am convinced of this paper’s quality. I recommend an accept, good work.

---

### Decision · Program_Chairs · 2022-04-17

Accept